# The Role of Hepatic Fat Accumulation in Glucose and Insulin Homeostasis—Dysregulation by the Liver

**DOI:** 10.3390/jcm10030390

**Published:** 2021-01-20

**Authors:** Amalie London, Anne-Marie Lundsgaard, Bente Kiens, Kirstine Nyvold Bojsen-Møller

**Affiliations:** 1Department of Endocrinology, Copenhagen University Hospital Hvidovre, Kettegård Allé 30, 2650 Hvidovre, Denmark; kirstine.nyvold.bojsen-moeller@regionh.dk; 2Section of Molecular Physiology, Department of Nutrition, Exercise and Sports, University of Copenhagen, Universitetsparken 13, 2100 Copenhagen, Denmark; amlundsgaard@nexs.ku.dk (A.-M.L.); bkiens@nexs.ku.dk (B.K.)

**Keywords:** non-alcoholic fatty liver disease (NAFLD), glucose metabolism, insulin resistance, hepatic glucose production, hyperinsulinemic-euglycemic clamp, proton MR spectroscopy, insulin clearance, molecular regulation

## Abstract

Accumulation of hepatic triacylglycerol (TG) is associated with obesity and metabolic syndrome, which are important pathogenic factors in the development of type 2 diabetes. In this narrative review, we summarize the effects of hepatic TG accumulation on hepatic glucose and insulin metabolism and the underlying molecular regulation in order to highlight the importance of hepatic TG accumulation for whole-body glucose metabolism. We find that liver fat accumulation is closely linked to impaired insulin-mediated suppression of hepatic glucose production and reduced hepatic insulin clearance. The resulting systemic hyperinsulinemia has a major impact on whole-body glucose metabolism and may be an important pathogenic step in the development of type 2 diabetes.

## 1. Introduction

Obesity is a central component in the development of metabolic syndrome [1,2], which is characterized and defined by central obesity (i.e., increased waist circumference) or body mass index (BMI, weight (kg)/height (m)^2^) > 30 combined with two of the following: elevated circulating triacylglycerol (TG), reduced high-density lipoprotein (HDL) cholesterol, hypertension and elevated fasting plasma glucose [3]. Although not included as a parameter in metabolic syndrome, insulin resistance is considered a core pathogenic factor in the development of metabolic syndrome, which is often associated with systemic hyperinsulinemia [4]. Central obesity, determined by the waist to hip ratio or waist circumference, is closely associated with excess TG accumulation in the liver [4,5,6,7] and with increased visceral fat content [8,9]. Notably, liver TG accumulation, rather than visceral fat, is suggested to be the primary driver for the above-mentioned metabolic derangements associated with central obesity and MS [10,11]. This is supported by studies of obese individuals where higher total body adiposity, including doubling of visceral fat mass, does not lead to additional metabolic derangements if hepatic TG content is constant [12]. Provided that accumulation of hepatic TG is not attributable to consumption of alcohol, an excess TG content of > 55 mg/g liver (5.5%) is referred to as non-alcoholic fatty liver disease (NAFLD), which has an estimated prevalence of 25% in the general population [13,14]. In line with the importance of central obesity as opposed to total body weight, not all obese have NAFLD, but obesity is considered an important risk factor for NAFLD [15]. Thus, in an Italian population, 76% of the obese persons had NAFLD estimated by ultrasound compared with 16% in the non-obese [16].

Accumulation of hepatic TG can be quantified in liver biopsies, which is considered the gold reference standard for diagnosis but is inappropriate for repeated monitoring. Ultrasound and computer tomography (CT) imaging has limited accuracy, while magnetic resonance (MR) techniques assess hepatic TG more accurately by decomposing the liver signal into fat and water signal components [17]. In particular, quantification of hepatic TG by proton MR spectroscopy ([^1^H]-MRS) correlates well with the histological grading of hepatic TG [18,19] and allows for repeated measurements. [^1^H]-MRS, however, requires highly advanced equipment, only available at research departments, and trained staff.

In the following, we will summarize the effects of hepatic TG accumulation on hepatic glucose and insulin metabolism and the underlying molecular regulation in order to highlight the importance of hepatic TG accumulation for whole-body glucose metabolism.

## 2. Liver TG Accumulation and Hepatic Glucose Production and Insulin Resistance

Hepatic glucose production (HGP) is regulated by several direct and indirect pathways involving hormones, neural regulation and availability of substrates [20]. Importantly, HGP is highly regulated by insulin, which suppresses the production of glucose by the liver. Insulin resistance of the liver is therefore characterized by elevated HGP in the basal fasting state and impaired suppression of HGP in the postprandial state. The gold standard for estimation of whole-body insulin sensitivity is the hyperinsulinemic-euglycemic clamp, which can be combined with the prior and concomitant infusion of a glucose tracer to estimate basal endogenous glucose production and the insulin-mediated suppression of endogenous glucose production during the clamp, respectively [21]. Since most of the endogenous glucose production originates from the liver with only a minor contribution from the kidneys, endogenous glucose production and HGP are used synonymously. To allow for estimation of the suppression of hepatic glucose production during the hyperinsulinemic-euglycemic clamp, lower infusion rates of insulin are typically needed compared with studies aiming at evaluating skeletal muscle insulin sensitivity [22]. When HGP is only measured in the basal condition, the hepatic insulin sensitivity index (HISI) can be calculated as the inverse of the product of basal HGP and plasma insulin concentrations to provide a surrogate index of hepatic insulin sensitivity [23]. Another often used surrogate index of hepatic insulin resistance (i.e., the reverse of insulin sensitivity) is the homeostasis model assessment of insulin resistance (HOMA-IR) calculated from plasma glucose and insulin concentrations in the fasting state [23,24,25]. Given the large extraction of insulin, but not C-peptide, by the liver [19], it seems necessary to use plasma C-peptide rather than plasma insulin concentrations for HISI as well as HOMA-IR, as C-peptide is a more reliable measurement of pre-hepatic insulin concentrations. This is, however, seldomly used.

When HGP was evaluated during a low-dose euglycemic insulin clamp (insulin infusion: 0.3 mU/kg/min or 12 mU/m^2^ body surface area (BSA)/min) in overweight glucose tolerant individuals (BMI 25–28 kg/m^2^), grouped according to high or low hepatic TG content (15 vs. 2%), the suppression of HGP by insulin was lower in individuals with high hepatic TG content whereas basal HGP was similar in a context of higher basal C-peptide and insulin concentrations (e.g., lower HISI) [19]. Interestingly, the percent suppression of HGP during the clamp correlated negatively with liver TG (*r*= −0.4) [19]. Similar results have been reported in individuals with near-normal BMI (25–26 kg/m^2^), where a high liver TG content (10.5 vs. 1.7%) was associated with less suppression of HGP (55 vs. 85%) during a low-dose insulin clamp (0.3 mU/kg/min) and comparable basal HGP in the context of high basal C-peptide and insulin concentrations [26]. In glucose tolerant obese individuals with a wide range of hepatic TG (1–46%), an inverse linear relationship between accumulation of hepatic TG and basal hepatic insulin sensitivity estimated by HISI was shown, which may be driven by a positive correlation between hepatic TG and basal insulin concentrations [27]. The link between high hepatic TG and low HISI is independent of BMI and visceral fat content determined by MR [10]. In line with this, individuals with equally high liver TG (mean of 14%), despite marked differences in total adiposity (BMI 41 vs. 31 kg/m^2^) and visceral fat content (two-fold difference), had similar impaired hepatic insulin sensitivity, measured by HISI and insulin-mediated suppression of HGP (20 mU insulin/m^2^ BSA/min) [12]. In a large US cohort of 352 individuals, impaired suppression of HGP during a low-dose insulin clamp (10 mU/m^2^ BSA/min) was observed already in individuals with hepatic TG exceeding 1.5%, whereas the basal insulin concentration did not demonstrate a threshold, but increased progressively with higher hepatic TG resulting in progressively increasing HOMA-IR [28].

In contrast to the normal glucose tolerant subjects, individuals with type 2 diabetes are characterized by overt elevated basal HGP despite high basal plasma insulin or C-peptide concentrations [22,29]. In a study of lean and obese individuals with type 2 diabetes, basal HGP did not correlate with hepatic TG content, whereas HISI and the suppression of HGP during a low dose insulin infusion (240 pmol/min/m^2^) correlated negatively with hepatic TG [29], supporting that hepatic TG is also negatively correlated with hepatic insulin sensitivity in individuals with type 2 diabetes. In obese NAFLD individuals without known diabetes, abnormal glucose metabolism, including prediabetes or overt type 2 diabetes (determined by oral glucose tolerance test, OGTT) was found in 85% of individuals vs. 30% in a control overweight population matched on total body fat [30]. In this population, HISI was approximately 50% lower in the presence of NAFLD independently of glucose tolerance, whereas the suppression of HGP during a low-dose insulin clamp (10 mU/m^2^ BSA/min) was more impaired in NAFLD individuals with prediabetes/diabetes (−44%) compared with individuals without NAFLD (−61%). Conversely, hepatic TG is 4-fold higher in individuals with than without metabolic syndrome independently of BMI [4], a finding that is supported by others [7].

Interestingly, with certain genetic forms of liver steatosis, the link between hepatic TG and hepatic insulin resistance may not be as clear. In familial hypobetalipoproteinemia, characterized by dysfunctional apolipoprotein B, which consequently causes reduced export of TG from the liver, affected children with liver steatosis have markedly lower HOMA-IR compared with children with NAFLD [31]. However, when compared with healthy controls, adults with familial hypobetalipoproteinemia tended to have higher HOMA-IR in a small study (*n* = 7) [32]. For the polymorphisms in adiponutrin/patatin-like phospholipase-3 (PNPLA3) and transmembrane 6 superfamily member 2 protein (TM6SF2), which are closely associated with liver steatosis, results are conflicting, but with several studies supporting a dissociation between liver fat content and insulin resistance [33,34,35].

In summary, accumulation of hepatic TG is negatively correlated to hepatic insulin sensitivity determined as impaired insulin-mediated suppression of HGP both in individuals with and without type 2 diabetes, supporting that accumulation of liver fat in itself could be responsible for impaired insulin action in the liver. Moreover, the studies support that accumulation of hepatic TG is closely linked to the metabolic derangements observed in metabolic syndrome and type 2 diabetes, although the link may be less clear in genetic variants of NAFLD.

### 2.1. Mechanisms in Increased Hepatic Glucose Production

The impaired action of insulin in the steatotic liver appears specific to the effects on HGP [36], but the underlying mechanisms are not fully understood, likely due to the complexity of regulation of HGP influenced (directly and indirectly) by multiple hormonal and neural inputs, as well as fluctuations in hepatic substrate availability [20,37].

Several studies have shown that in individuals with hepatic steatosis with and without type 2 diabetes, HGP is associated with increased hepatic gluconeogenesis rather than increased glycogenolysis when compared with lean, overweight or obese individuals with normal liver TG [29,38,39]. The increased gluconeogenesis has been shown to originate from oxaloacetate (from the pyruvate carboxylase reaction) [39], with possible contributions from pyruvate, lactate and amino acids [40], while gluconeogenesis from glycerol, on the other hand, seems to be diminished in liver steatosis [41].

### 2.2. Liver TG Accumulation and Hepatic Insulin Signaling

Insulin signaling in the liver is effectuated by insulin binding to the hepatic insulin receptor (IR) and downstream activation of insulin receptor substrate (IRS) and phosphoinositide 3-kinase (PI3-kinase), activating Akt that acutely activates glycogen synthase and inactivates forkhead box-containing protein O subfamily 1 (FOXO1). Insulin binding thereby results in glycogen synthase activation and transcriptional downregulation of gluconeogenic enzymes, the latter mediated by FOXO1 nuclear export [42]. RNA sequencing has demonstrated overall downregulation of insulin signaling genes in the liver of obese individuals with NAFLD and non-alcoholic steatohepatitis (NASH) liver samples compared with lean and obese controls without steatosis [43].

In liver biopsies from individuals with NAFLD, gene expression analyses revealed a lower ratio between the IR isoforms type A and B, with potential implications for downstream insulin signaling, and lower IRS-2 mRNA expression compared with individuals with normal liver TG [44,45], while gluconeogenic enzymes as glucose-6-phosphatase (G6Pase) and phosphoenolpyruvate carboxykinase (PEPCK) were increased [45,46]. Reduced Akt1/2 protein content and increased FOXO1 mRNA and nuclear localization, indices of increased FOXO1 activity, also has been demonstrated in the steatotic human liver [46].

The link between increased liver TG content and impaired insulin signaling is not known. It has been shown that in severely obese individuals with steatosis, the hepatic content of diacylglycerol (DAG) correlated positively with liver TG content and negatively with insulin-mediated suppression of glucose production [47], and it was shown in obese individuals that lipid-droplet associated DAG in liver samples correlated with HOMA-IR, and with protein kinase C ε (PKCε) activation, the main PKC isoform in human liver, which impairs insulin signaling [48]. Moreover, increased cytosolic DAG content and increased PKCƐ activation was obtained in liver samples from individuals with impaired versus normal insulin suppression of HGP [49]. These studies support a mechanistic model in which increasing TG and DAG accumulation in the liver activates PKC, thereby impairing insulin signaling, which adds to the findings of a more general transcriptional downregulation of insulin signaling shown by others [43]. Evidence from human studies on the role of ceramides in the insulin resistance of the steatotic liver is sparse. In individuals with NAFLD, ceramide content was not increased (while DAG was increased) [50], and there were no apparent associations between liver ceramide content and insulin sensitivity or HOMA-IR in any of the associative studies [47,48,49].

### 2.3. Increased Gluconeogenic Flux

Hepatic tricarboxylic acid (TCA) flux is positively correlated with the degree of steatosis in the human liver [51]. By the use of stable propionate tracers and MR Spectroscopy in obese individuals with liver steatosis, the TCA cycle flux and pyruvate carboxylase flux (conversion of pyruvate to oxaloacetate) were elevated when compared with obese controls without steatosis [38,39]. Thus, there seems to be an increased oxidative metabolism in the steatotic liver, which leads to increased oxaloacetate-driven gluconeogenesis due to increased pyruvate cycling (the main pathway in mitochondrial anaplerosis) (Figure 1). Elevated TCA flux and pyruvate cycling has thus been speculated to function as a progenitor of the increased gluconeogenesis, which may represent an important mechanism in the elevated glucose production in NAFLD.

Pyruvate represents the upstream precursor of oxaloacetate and several lines of evidence point to a role of increased pyruvate availability in hepatic TG accumulation. Findings from a metabolomics study of arterial-hepatic venous blood sampling showed elevated mitochondrial pyruvate transport and flux through pyruvate carboxylase in individuals with high versus normal liver TG [40]. Furthermore, a recent study showed an increased gene expression of pyruvate kinase (PK), catalyzing the final step in glycolysis, yielding pyruvate, in liver biopsies from men with steatosis compared with normal liver fat [52]. Moreover, the gene and protein expression of pyruvate dehydrogenase kinase 4 (PDK4), which is a negative regulator of pyruvate dehydrogenase (PDH) activity, was shown to be higher in the liver biopsies of individuals with NAFLD [53]. This would result in less pyruvate conversion to acetyl coenzyme A (acetyl-CoA), and hence increased substrate for the pyruvate carboxylase reaction and hence oxaloacetate substrate for gluconeogenesis. In contrast, if the glycolytic flux is suppressed, this would lead to lower pyruvate-driven gluconeogenesis.

## 3. Liver TG Accumulation and Systemic Hyperinsulinemia: The Effects on Insulin Clearance

Impaired action of insulin in the liver, as well as in muscle and adipose tissue, is central in the pathogenesis of type 2 diabetes, which is typically perceived to develop when the compensatory systemic hyperinsulinemia is no longer sufficient to overcome the progressive increase in insulin resistance [21,54]. The compensatory increase in circulating insulin concentration has been attributed to an increase in pancreatic insulin secretion, but changes in hepatic insulin clearance are also contributing to the systemic hyperinsulinemia and may even be of greater importance, although widely debated [55,56,57].

The liver is the main organ for insulin clearance and, to a lesser extent, kidneys, muscle and adipose tissue [58,59]. It has been estimated from insulin infusion studies that approximately 60–70% of insulin is cleared in the liver in the healthy non-diabetic, and non-steatotic state [60,61], a fractional extraction which may be even higher for endogenous insulin given the first-pass metabolism of portal insulin [62]. Insulin is secreted from pancreatic beta-cells equimolarly with C-peptide, but unlike insulin, C-peptide is not extracted by the liver [63]. The ratio of C-peptide to insulin concentrations can therefore under steady-state conditions, as in the fasted state, provide an indirect estimate of hepatic insulin clearance, whereas the use of the C-peptide to insulin ratio during nonsteady state conditions requires integrated measurements such as Area-Under-the-Curves (AUCs) provided that both C-peptide and insulin concentrations have returned to basal levels [58,63,64]. Whole-body insulin clearance can be estimated using the insulin clamp technique by dividing insulin infusion rate by the mean steady-state plasma insulin with correction for C-peptide concentrations [65].

Hepatic TG accumulation is inversely correlated with whole-body insulin clearance during the hyperinsulinemic clamp in lean and obese individuals with and without type 2 diabetes [29]. Decreased insulin clearance contributes importantly to the high fasting plasma insulin concentrations in individuals with hepatic TG accumulation and normal glucose tolerance [19] as well as early-stage type 2 diabetes [66]. Hepatic insulin clearance, measured as the ratio between AUCs of C-peptide and insulin during an OGTT decreased linearly with increased hepatic TG content and explained the progressive increase in basal insulin concentration and HOMA-IR in the large US cohort of 352 middle-aged obese individuals (mean BMI, 33.1 ± 5.3 kg/m^2^) including a 61% prevalence of type 2 diabetes [28]. In fact, of all components of the metabolic syndrome, systemic insulin concentrations in the fasting state had the strongest correlation with hepatic TG [4]. In line with this, insulin clearance measured during the hyperinsulinemic clamp was significantly decreased in metabolically abnormal obese individuals compared with metabolically healthy obese individuals with similar BMI [67]. In the above-mentioned study, individuals were classified as metabolically abnormal obese if they were characterized with insulin resistance, measured with the insulin clamp, elevated plasma TG, low plasma HDL concentrations, elevated blood pressure or impaired fasting glucose [67]. Moreover, postprandial hyperinsulinemia during an OGTT in obese NAFLD individuals has been shown to strongly correlate with reduced hepatic insulin clearance but not insulin secretion [68]. In a large cross-sectional study including 532 obese young individuals, it was shown that insulin clearance and the hepatic insulin resistance index calculated during an OGTT was ~50% lower in individuals with steatosis [69].

In summary, hepatic insulin clearance is negatively associated with hepatic TG content and is an important determinator of systemic hyperinsulinemia seen in conditions characterized by excess hepatic TG accumulation. The close association between hepatic TG and insulin clearance is important to recognize when estimating insulin sensitivity and insulin secretion in groups differing in hepatic TG or in response to interventions affecting hepatic TG content (i.e., diet, bariatric surgery, pharmacological interventions). Indices of insulin sensitivity or insulin resistance determined by fasting insulin concentrations such as HOMA-IR will potentially overestimate whole-body insulin resistance in the context of excess liver TG and impaired insulin clearance [28,70]. The use of C-peptide instead of insulin concentrations, which can be applied in the HOMA2-model [71], is likely to reduce this issue. Similarly, when evaluating the beta-cell function, it is crucial to use C-peptide rather than insulin concentrations since the reduced insulin clearance associated with liver TG accumulation will lead to an overestimation of insulin secretion and vice versa in the context of low hepatic TG.

### Mechanisms in Hepatic Insulin Clearance

Hepatic insulin clearance involves insulin binding to the insulin receptor on hepatocytes [72] and is therefore closely linked to hepatic insulin action. When it comes to insulin clearance by the liver, regulation seems to reside at the level of insulin endocytosis and insulin degradation. The protein carcinoembryonic antigen-related cell adhesion molecule 1 (CEACAM1) promotes internalization of the insulin-insulin-receptor complex, as shown in vitro [73], by insulin receptor tyrosine kinase-induced phosphorylation of CEACAM1 [74], which subsequently forms part of a protein complex mediating insulin-receptor endocytosis [75]. It is believed that insulin is then degraded by lysosomal proteolysis in the hepatocytes, by the actions of an insulin-degrading enzyme (IDE). IDE mRNA expression is lower in liver samples of individuals with type 2 diabetes and NAFLD compared with healthy individuals with normal liver TG [75]. A low CEACAM1 activity might thus lead to reduced insulin receptor endocytosis. In accordance, liver-specific CEACAM1 deletion in mice leads to impaired insulin clearance, systemic hyperinsulinemia, and impaired hepatic insulin sensitivity [76]. Long-chain fatty acids provide ligands for the peroxisome proliferator-activated receptors (PPARs) [77], of which PPARα is the dominant isoform in the human liver [78]. Activation of PPARα by pharmacological agonists has been shown to reduce CEACAM1 expression [72], and high-fat feeding in mice led to upregulation of PPARα and downregulation of hepatic CEACAM1 expression [73]. PPARα activation could thus represent a link between elevated fatty acid availability in the liver and impaired insulin clearance. In support of this, hepatic CEACAM1 protein expression was recently found to be reduced in obese compared with lean individuals [79], and in individuals with NAFLD [80]. Insulin clearance may thus, via CEACAM1 expression and activity, play an important role in determining insulin action in the liver.

## 4. The Importance of Hepatic Insulin Clearance for Whole-Body Glucose Homeostasis

Liver TG accumulation and the associated reduction in hepatic insulin clearance results in increased insulin availability to the peripheral tissues. Reduced hepatic insulin clearance could thus represent an initial homeostatic mechanism to preserve β-cell function in the context of increased whole-body insulin resistance [81]. However, the systemic hyperinsulinemia caused by reduced hepatic insulin clearance may also act as a stressor of peripheral tissues perturbating peripheral insulin sensitivity [55]. A recent study in adolescents demonstrated a close association between insulin clearance and peripheral insulin sensitivity measured by glucose disposal during a hyperinsulinemic-euglycemic clamp [82], which, however, does not clarify whether reduced hepatic insulin clearance is a cause or consequence of impaired peripheral insulin sensitivity. Interestingly, the study followed participants for 2 years and demonstrated that low insulin clearance was the sole predictor of impaired beta-cell function at the follow-up independently of baseline beta-cell function and BMI [82]. In addition, in a longitudinal 9-year study, lower baseline insulin clearance and decline in insulin clearance measured as the C-peptide to insulin ratio during an OGTT were related to the incidence of prediabetes and diabetes [83]. In line with this, when oral glucose tolerance was assessed in 75 young overweight men with steatosis, HOMA-IR was the most important predictor (Odds Ratio of 3.4 compared with no steatosis) of impaired glucose tolerance, with the glucose-intolerant individuals characterized by 70% higher fasting insulin levels [84], likely related to reduced insulin clearance.

Impaired insulin clearance may therefore be an early sign of metabolic derangement, highlighting the need for early intervention of liver fat accumulation to obtain normalization of hepatic insulin metabolism and prevention of type 2 diabetes. The causal link between hepatic fat accumulation and impaired hepatic gluco-regulation, insulin action and insulin clearance and the impact on whole-body metabolism requires further studies, including elucidation of molecular mechanisms. Of interest is the dynamic response of liver TG and the associated impairments in liver glucose and insulin metabolism, where studies support rapid reversals in response to initiation of acute caloric restriction. Hence after gastric bypass surgery, reductions in hepatic TG and improvements in hepatic insulin sensitivity measured by HISI (using C-peptide) are observed within days before major weight loss has occurred [85,86], underscoring that the postoperative caloric restriction rapidly lowers hepatic TG and improves hepatic insulin sensitivity. Concomitantly, insulin clearance, measured during a hyperinsulinemic-euglycemic clamp and with the C-peptide-to-insulin ratio, is increased within days after RYGB [85,87]. Similarly, calorie restriction induced by very-low-calorie diets (600–1400 kcal/day) reduces hepatic TG content and lowers basal HGP in obese individuals with and without type 2 diabetes within days to weeks [88,89,90]. In line with this observation, even a modest diet induced weight loss (<10% of initial body weight) leads to a marked reduction in hepatic TG (by 60–80%) and concomitant improvements in hepatic insulin sensitivity measured by HISI and insulin-mediated suppression of HGP [91,92].

Hepatic TG accumulation also seems to be strongly affected by modification of dietary composition. In obese individuals, carbohydrate restriction was shown to be superior to fat restriction under hypocaloric conditions (deficit of 1000 calories/day), resulting in a more pronounced decrease in hepatic TG (−30%) in the group consuming a low carbohydrate diet (<60 g/day) compared with the high carbohydrate group (180 g/day) (−10%) [90]. Notably, the effects were observed after a remarkably short intervention period of only 48 h [90]. Carbohydrate restriction may even be efficient during eucaloric conditions, as recently shown in obese individuals with NAFLD, where initiation of a eucaloric low-carbohydrate diet (<30 g/day) resulted in a significant reduction of hepatic TG within one day and a mean reduction of 43.8% in hepatic TG content after 14 days [93]. Although the diet was aimed to be eucaloric, a minor weight loss was observed (1.8 ± 0.2% of total body weight). Other studies support lowering of hepatic TG after intake of eucaloric low carbohydrate diet (30% energy (E%)) compared with a control diet after six weeks in patients with type 2 diabetes [94]. Conversely, three days of overfeeding with carbohydrates (and not fat) reduce insulin clearance measured by the hyperinsulinemic-euglycemic clamp simultaneous with an increase in basal HGP supporting impairments in hepatic insulin sensitivity after short-term excess carbohydrate intake [95]. These dynamic and major impacts on liver fat content in response to short-term dietary changes underscore the importance of controlling diet in the days prior to estimation of hepatic fat content.

## 5. Conclusions

In conclusion, liver fat accumulation is closely linked to impaired insulin-mediated suppression of hepatic glucose production and reduced hepatic insulin clearance. The resulting systemic hyperinsulinemia has major impacts on whole-body glucose metabolism and may be an important pathogenic factor in the development of type 2 diabetes. Moreover, the liver is a dynamic organ that rapidly adapts to changes in macronutrient and energy availability within hours or days by adjusting liver fat content, which is associated with marked changes in hepatic glucose metabolism and insulin clearance. A more detailed molecular understanding of how the accumulation of liver TG interferes with glucose regulation and insulin sensitivity and clearance could reveal potential treatment targets to address important metabolic derangements associated with obesity.

## Figures and Tables

**Figure 1 jcm-10-00390-f001:**
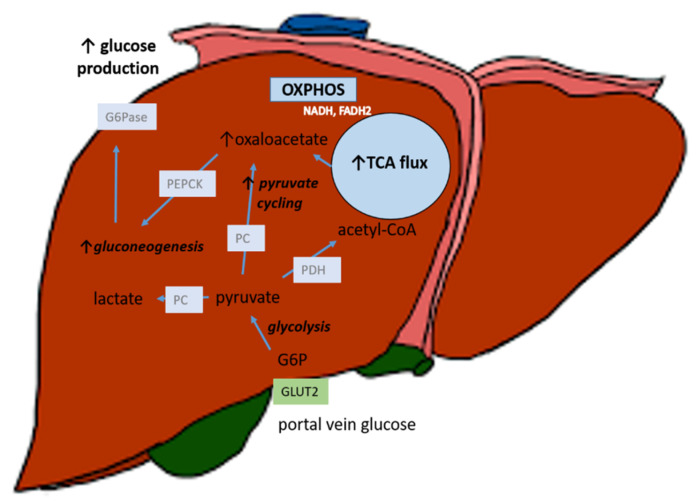
In the steatotic liver, increased mitochondrial tricarboxylic acid cycle (TCA) flux and increased pyruvate cycling (anaplerosis) potentially lead to increased oxaloacetate availability for gluconeogenesis. GLUT2: glucose transporter 2. G6P: glucose-6-phosphate. PDH: pyruvate dehydrogenase. PC: pyruvate carboxylase. PEPCK: phosphoenolpyruvate carboxykinase. OXPHOS: oxidative phosphorylation.

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
