# Peer review of "The Role of Hepatic Fat Accumulation in Glucose and Insulin Homeostasis—Dysregulation by the Liver"

_jcm, 2021, doi:10.3390/jcm10030390_

Round 1

Reviewer 1 Report

Review of manuscript No. JCM-1071569: „ The role of hepatic fat accumulation in glucose and insulin homeostasis – dysregulation by the liver”, written by Amalie London, Anne-Marie Lundsgaard, Bente Kiens and Kirstine Nyvold Bojsen-Møller, which will be published in Journal of Clinical Medicine.

The structure of the manuscript is in keeping with the standard required criteria. The topic of the work is very actual. Obesity is a central component in the development of metabolic syndrome, which is characterized and defined by central obesity (increased waist circumference) or body mass index (BMI, weight (kg)/ height (m)2) >30. Central obesity, determined by the waist to hip ratio or waist circumference, is closely associated with excess TG accumulation in the liver and increased visceral fat content.

Liver TG accumulation and the associated reduction in hepatic insulin clearance results in increased insulin availability to the peripheral tissues. Reduced hepatic insulin clearance could thus represent an initial homeostatic mechanism to preserve β-cell function in the context of increased whole-body insulin resistance. However, the systemic hyperinsulinemia caused by reduced hepatic insulin clearance may also act as a stressor of peripheral tissues perturbating peripheral insulin sensitivity. A recent study in adolescents demonstrated a close association between insulin clearance and peripheral insulin sensitivity measured by glucose disposal during a hyper-insulinemic euglycemic clamp.

In conclusion, the authors identified that liver fat accumulation is closely linked to impaired insulin-mediated suppression of hepatic glucose production and reduced hepatic insulin clearance. The resulting systemic hyperinsulinemia has significant impacts on whole-body glucose metabolism and maybe a critical pathogenic factor in developing type 2 diabetes.

This paper is readable, brings summarizes new knowledge. The citations are actual, and their format respects the usual standards.

I recommend the manuscript to be published.

Košice, 6. January 2021

MUDr. Jana Katuchova, PhD.

Professor of Department of Surgery

Medical Faculty of Safarik University and

University Hospital Košice

Slovakia

Author Response

Response to Reviewers

We sincerely thank you for your constrictive comments, which helped to improve the manuscript. Please find point-by-point responses to the reviewers’ comments below.

Reviewer 1#: Review of manuscript No. JCM-1071569: „ The role of hepatic fat accumulation in glucose and insulin homeostasis – dysregulation by the liver”, written by Amalie London, Anne-Marie Lundsgaard, Bente Kiens and Kirstine Nyvold Bojsen-Møller, which will be published in Journal of Clinical Medicine.

The structure of the manuscript is in keeping with the standard required criteria. The topic of the work is very actual. Obesity is a central component in the development of metabolic syndrome, which is characterized and defined by central obesity (increased waist circumference) or body mass index (BMI, weight (kg)/ height (m)2) >30. Central obesity, determined by the waist to hip ratio or waist circumference, is closely associated with excess TG accumulation in the liver and increased visceral fat content.

Liver TG accumulation and the associated reduction in hepatic insulin clearance results in increased insulin availability to the peripheral tissues. Reduced hepatic insulin clearance could thus represent an initial homeostatic mechanism to preserve β-cell function in the context of increased whole-body insulin resistance. However, the systemic hyperinsulinemia caused by reduced hepatic insulin clearance may also act as a stressor of peripheral tissues perturbating peripheral insulin sensitivity. A recent study in adolescents demonstrated a close association between insulin clearance and peripheral insulin sensitivity measured by glucose disposal during a hyper-insulinemic euglycemic clamp.

In conclusion, the authors identified that liver fat accumulation is closely linked to impaired insulin-mediated suppression of hepatic glucose production and reduced hepatic insulin clearance. The resulting systemic hyperinsulinemia has significant impacts on whole-body glucose metabolism and maybe a critical pathogenic factor in developing type 2 diabetes.

This paper is readable, brings summarizes new knowledge. The citations are actual, and their format respects the usual standards.

I recommend the manuscript to be published.

We thank the reviewer for the positive response and comments and for reviewing the manuscript.

Reviewer 2 Report

The revision article entitled “The role of hepatic fat accumulation in glucose and insulin ho-meostasis – dysregulation by the liver” by Amalie London and co-workers aimed to analyze the link between hepatic TG accumulation and whole-body glucose metabolism. The manuscript is interesting and significantly contributes to a better understanding of the molecular mechanisms beyond excessive hepatic TG accumulation and glucose metabolism.

In page 2, the first paragraph regarding fatty liver diagnosis is a little bit out of place. Maybe this paragraph could be relocated in the Introduction section.

In the second paragraph of page 3, the authors indicate that in the study of Ortiz-López and co-workers, overweight and obese patients with NAFLD had an increased HISI comparing to those without NAFLD. However, in their article the authors stated that participants with NAFLD present higher fasting hepatic insulin resistance (HIR) than those without NAFLD (figure 2A, Ortiz-López et al.). The authors should check the study to confirm or modify their statement.

At the end of the first paragraph of “Increased gluconeogenic flux” section on page 4, the authors indicate that in steatotic individuals there is a higher TCA flux that could be given to an increased oxalacetate production. Taken into account that oxalacetate is an important gluconeogenic substrate, I suppose that the authors suggest that this “excess” of oxalacetate production could be directed to glucose production (gluconeogenesis). If this supposition is true, the authors should state more clearly to a better understanding.

Also, in the same section, quite clearly written but complex explanations about the relationship among steatosis, TCA flux (pyruvate and oxalacetate) and gluconeogenesis are included. The authors should consider illustrating those explanations in a figure. In addition, a summatory sentence indicating the potential fact that all those metabolic pathway changes result in an increased gluconeogenesis could be added.

In the third paragraph of page 5 (about reference 28), the authors refer to study participants as overweight individuals being their mean BMI 33.1. According to WHO classification, these participants are obese.

In the same paragraph: “In line with this, insulin clearance measured during the hyperinsulinemic clamp was signifi-cantly decreased in obese metabolically, unhealthy individuals compared with metaboli-cally, healthy individuals with similar BMI [61]”. Taking into account that the “metabolically unhealthy/unhealthy” term has not been mentioned so far, and that this term is not standardised defined, the authors should include briefly how are obese participants classified as metabolically healthy or unhealthy (according to the cited article). For example, metabolically healthy obese when they present lower total cholesterol, insulin resistance...  

Page 6, near the end of the paragraph: PPARα-activation reduces CEACAM1 expression. Why this fact is responsible for the link between lipid bioavailability in the liver and insulin clearance? Ii is because lipids are PPARα-ligands? That being the case, it should be included in the text.

In my humble opinion, the “Shift towards a metabolic healthy liver: effects of interventions reducing liver fat” section is slightly out of place in this review article, especially bearing in mind that the aim of the review is “the effects of hepatic TG accumulation on he-patic glucose and insulin metabolism and the underlying molecular regulation in order to highlight the importance of hepatic TG accumulation for whole-body glucose metabo-lism.”. Even more, this fact is not included in the Conclusion. Anyhow, in the case of including this section, I recommend putting it at the end of the article; the fact that the next section is about “The role of the liver for whole body glucose metabolism” is peculiar for me.

In the abovementiones section, the authors claim that low-carbohydrate diets are the most effective in reducing hepatic TG accumulation. In order to reinforce this statement, the authors state that 3 days of overfeeding with carbohydrates reduces insulin clearance. I am not sure about the adequacy of this comparison, due to carbohydrate excess it is not usually used as “anti-obesity” approach (the goal of this section is obesity and co-morbidities treatment approaches).

The authors entitled the last section as “The role of the liver for whole body glucose metabolism”. However, little is about this issue. The authors should change it, or even redesign these final paragraphs. Is the length of this section until the last paragraph (conclusion)? Is the penultimate paragraph included in this section, or just final reflections? I encourage the authors to rethink the end of the article.

Minor comments:

  • “Introduction” or similar headline should be included at the beginning of the manuscript
  • First paragraph: it seems that there is a sentence in bold font
  • Pages 2-3: “In line with this, two groups of individuals with equally high liver TG (mean of 14%) had similar impaired hepatic insulin sensitivity, measured by HISI and insulin mediated sup-pression of HGP (20 mU insulin/m2 BSA /min) and similar basal plasma insulin concen-trations, despite marked differences in obesity (BMI 41 vs 31 kg/m2) and 2-fold greater visceral fat content in the morbidly obese group [12]”.This sentence is too long and a little bit difficult to follow. Could you put in a more simpler way?
  • The second sentence in the third paragraph of page 4 is really long and too much concept have been included. Maybe it could be fragmented and better explained for an easier understanding.
  • Reference 56 in the text: a space must be included between “insulin” and “[56]”.
  • Page 5, third paragraph, first sentence: please divide this long sentence, for example writing the first one until reference 29.
  • Page 5, third paragraph: “In fact, of all components of the metabolic syndrome systemic insulin con-centrations in the fasting state had the strongest correlation with hepatic TG”. To a better understanding, a comma after syndrome may be included.
  • Conclusion should be written in “Conclusion” entitled section (in a separate section)
  • Last paragraph: impairired = impaired

Author Response

Response to Reviewers

We sincerely thank you for your constrictive comments, which helped to improve the manuscript. Please find point-by-point responses to the reviewers’ comments attached.

Reviewer 3 Report

The paper by London and colleagues reviews the available evidence on insulin homeostasis in relation to fatty liver disease. This is not a systematic review, but rather a revision of the evidence supporting the concept that increased fatty liver content is associated to insulin resistance. The authors have written an interesting paper, on a subject that is increasingly interesting due to the high prevalence of fatty liver disease in the world. The authors suggest that fatty liver may be associated to decreased insulin clearance, being this a physiological response of the system to a noxious signal.  However, the authors fail to provide evidence that may contradict their view. As it is, the paper is not well balanced. At least the authors should make more efforts to provide that type of information. Consequently, the authors need to discuss situations in which the liver fat content may be high, but without insulin resistance. One of the best examples is for example hypobetaliproteinemia, a disorder in which the liver accumulates fat because there is a defect in the secretion of VLDL due to a truncated apoB100 protein (see for example: Della Corte et al. Clin Endocrinol  2013; 79: 49-54). Another example is abetalipoproteinemia in which there is a defect in the MTTP molecule and therefore lipidation of the nascent VLDL particle is impeded resulting intracellular degradation and accumulation of fat in the liver cell. As far as we know, these situations have not been linked to increased insulin resistance or decreased insulin clearance.  Some evidence may also be derived from the clinical experiments using MTTP-inhibitors.

Another issue that remains unclear is the cause-relationship between fatty liver and insulin resistance, although this may be a difficult one.

In the paragraph on DAG accumulation, the authors may want to consider some comments on ceramides in this respect?

Some typos:

Page 4, second paragraph of “Incrteased gluconeogenic flux”: “points” should read “point”

Page 7, last paragraph: impaired

Author Response

(The authors gave the same response as above.)
